# Electrochemical Behaviors of Ni-Base Superalloy CMSX-4 in 3.5 wt.% NaCl: Effect of Temperature and Preoxidation

**DOI:** 10.3390/ma13235478

**Published:** 2020-12-01

**Authors:** Xianzi Lv, Quantong Jiang, Jie Zhang, Jianxin Zhang, Zaiwang Huang, Jizhou Duan, Baorong Hou

**Affiliations:** 1Key Laboratory of Marine Environmental Corrosion and Bio-fouling, Institute of Oceanology, Chinese Academy of Sciences, No.7 Nanhai Road, Qingdao 266071, China; lvxianzi@qdio.ac.cn (X.L.); jiangquantong@qdio.ac.cn (Q.J.); duanjz@qdio.ac.cn (J.D.); brhou@qdio.ac.cn (B.H.); 2Open Studio for Marine Corrosion and Protection, Pilot National Laboratory for Marine Science and Technology, Qingdao 266237, China; 3Center for Ocean Mega-Science, Chinese Academy of Sciences, 7 Nanhai Road, Qingdao 266071, China; 4Key Laboratory for Liquid-Solid Structural Evolution & Processing of Materials (Ministry of Education), Shandong University, Jinan 250061, China; jianxin@sdu.edu.cn; 5State Key Laboratory of Powder Metallurgy, Central South University, Changsha 410083, China; huangzaiwang@csu.edu.cn; 6Powder Metallurgy Research Institute, Central South University, Changsha 410083, China

**Keywords:** electrochemical corrosion, Ni-base superalloy, corrosion products, SEM, preoxidation

## Abstract

The electrochemical behaviors of the Ni-base superalloy CMSX-4 were carried out in 3.5 wt.% NaCl solution using electrochemical technique. The electrochemical corrosion process was divided into four stages, and reactions at the alloy surface and corrosion morphology at each stage were analyzed. The passivity mechanism at the stable passivation stage and the occurrence of pitting corrosion at the transpassivation state were discussed especially. The corrosion parameters including *E*_corr_, *E*_pass_, *i*_pass_ and *E*_pit_ were compared at different temperatures to reveal the relationship between the temperature and the corrosion resistance properties. The corrosion products were investigated by the aid of X-Ray Diffraction (XRD) and Energy Dispersive Spectroscopy (EDS). By designing different preoxidation procedures, the corrosion mechanism of oxide scales was analyzed for the preoxidized samples.

## 1. Introduction

Ni-base superalloys have the ability to resist a wide variety of corrosive environments and loads over extended periods of time, becoming unique high-temperature materials used in turbine blades of industrial processes [1,2,3,4]. The CMSX-4 superalloy is a versatile alloy which has excellent corrosion resistance characteristics, and is widely applied in the fields of aircraft, marine engineering, and some military equipment [5,6,7]. In order to prolong the using life of the CMSX-4 superalloy, considerations should be given to its corrosion behaviors, together with their strong dependence on temperature and preoxidation procedures.

A variety of research on superalloys focus on the high-temperature oxidation and hot corrosion. The oxidation behavior of the single-crystal PWA 1438 at 950 °C in flowing air is characterized by the formation of a multi-layered oxide scale [8]. Oxidation of Co-base superalloys at high temperatures in air happens in two stages: a transient period and a subsequent steady-state period in which several oxides coexist in different layers [9]. The K52 alloy forms mainly chromia scale that exhibits poor adhesion to the substrate during oxidation in air at 800–1000 °C [10]. The high-temperature oxidations of superalloys show the formation of multi-layered oxides [9,10,11,12,13]. The K40S superalloy suffers from accelerated corrosion and forms non-protective layer with poor adhesion beneath the salt mixture deposit [14]. The corrosion products of the Ni-base superalloys by Na-Salts at high temperatures are porous and easily spalled with laminar structure [15,16]. The high-temperature oxidation and hot corrosion in traditional studies refer to chemical corrosion, failing to reflect the chemical and electrochemical reactions between the alloys and corrosive electrolyte under complex service environment. Recently, several groups tended to study the corrosion mechanism of nickel-based superalloys using electrochemical methods. Electrochemical dissolution behavior of Nickel-based superalloy revealed that the presence of M_23_C_6_ carbides in the grain boundaries lead to the formation of a number of preferred sites for micro-corrosion cracks at low current densities [17]. Electrochemical technique was also used to study the effect of the amount of P phase in microstructure on corrosion properties of the UNS N26455 Superalloy due to its selective dissolution [18]. Corrosion properties of the UNS N26455 Superalloy decreased as the TCP phase concentration increased in the microstructure [19].

There are two problems in the research on the corrosion behaviors of the CMSX-4 superalloy. (1) Though the CMSX-4 superalloy has high corrosion resistance, some issues including pitting corrosion [5,20], intergranular corrosion [21], and stress corrosion [7,22] are studied in a very few papers. Compared with general corrosion, local corrosion is more likely to happen and to induce local failure of superalloys. (2) Practically, the corrosion process of the CMSX-4 superalloy is an electrochemical process during service. Most papers about the electrochemical corrosion behaviors of the CMSX-4 superalloy focused on low temperature and pressure [23,24].

In this work, the electrochemical behaviors of the CMSX-4 superalloy in 3.5 wt.% NaCl solution were investigated to reveal its electrochemical corrosion process and the evolution from local pitting corrosion to general corrosion. The relationship between the corrosion resistance properties and the temperature was studied, together with the oxide films on the surface of the corroded samples. The influence of preoxidation on the electrochemical behaviors of the CMSX-4 superalloy was discussed to indicate the growth process and failure mechanism of oxide films.

## 2. Materials and Methods

### 2.1. Materials and SPECIMEN Preparation

The ingots of the alloys were prepared by vacuum induction melting (VIM) and the compositions were measured by inductively-coupled plasma-optical emission spectroscopy (ICP-OES) at NCS Testing Technology Co., Ltd., Changsha, China. Single crystal rods, with 13 mm in diameter and 170 mm in length, were directionally solidified using a conventional Bridgman method in a high rate solidification furnace (ALD furnace, Hanau, Germany). The melting temperature was 1500 °C and the withdrawal rate of the mold was 3.0 mm/min. The chemical composition of the chosen Ni-base superalloy CMSX-4 is Ni-9.0Co-6.5Cr-6.0W-6.5Ta-5.6Al-3.0Re-1.0Ti-0.6Mo-0.1Hf (wt.%). Specimens for electrochemical tests with a diameter of 13 mm and a thickness of 3 mm were machined from cast bars with their longitudinal axis parallel to the <001> direction. They have the same microstructural orientation. Before the tests, the surface of the samples was grinded with sandpaper (600#, 1000#, 2000#), then polished to the mirror with Al_2_O_3_ polishing paste (1.0 μm). Prior to the tests, specimens were rinsed with acetone, ethanol, and deionized water, and then dried in a drying oven.

### 2.2. Electrochemical Measurements

Electrochemical tests were performed using the aerated 3.5 wt.% NaCl solution which were prepared with deionized water. Electrochemical tests were carried out using a standard three-electrode system. The three-electrode system contained the specimen as working electrode, a saturated calomel electrode (SCE) reference electrode, and platinum counter electrode. In order to study the influence of temperatures on the electrochemical behaviors of the CMSX-4 superalloy, the anodic polarization tests were carried out using the Ag/AgCl electrode reference electrode in saturated KCl solution.

The specimens were immersed in the NaCl solution for 3 h to attain a relatively stable value of open-circuit potential (OCP). Potentiodynamic polarization tests were performed using the PARSTAT 4000+ (AMETEK, Princeton, NJ, USA). The potential swept from the OCP value towards the anodic direction at a scanning rate of 1.0 mV/s.

### 2.3. Specimen Characterization

The microstructure of the corroded specimens was examined by the S-3400N Scanning Electron Microscopy (SEM, HITACHI, Hitachi, Japan). The corrosion products were analyzed using Energy Dispersive Spectroscopy (EDS, mounted at the above SEM, HITACHI, Hitachi, Japan) and X-Ray Diffraction (XRD, Rigaku Cooperation, Tokyo, Japan, D/max-3C X-ray diffraction operating at 40 kV and 30 mA with Cu Kα radiation (λ = 0.15406 nm) from 10 °C to 80 °C).

## 3. Results and Discussion

### 3.1. Electrochemical Behavior of the CMSX-4 Superalloy

The anodic potentiodynamic polarization curves of the CMSX-4 superalloy in 3.5 wt.% NaCl solution at 20 °C, 50 °C and 80 °C are shown in Figure 1. The potential was applied on the working electrode at a scan rate of 0.167 mV/s starting from −0.1 mV to 1.0 V with respect to self-corrosion potential. The electrochemical corrosion process included active dissolution region, stable passivation region, transpassivation region, uniform oxygen evolution region.

The anodic polarization curve at 20 °C was taken for an example. The CMSX-4 superalloy had an OCP close to −285 mV (vs saturated Ag/AgCl), and as the potential became positive, the current density increased rapidly. Anodic dissolution of alloying elements was controlled by activation polarization [25]. The surfaces of the samples were smooth to the naked eye.

When the potential reached 198 mV, the value of current density was nearly 7.27 μA/cm^2^ and the surface of the samples was in a passive state. Then the current density was almost constant within a range of 198 mV to 331 mV, forming a passivation plateau. The formation of passive film layers for the CMSX-4 superalloy started with the formation of the two-dimensional membrane structure in which H_2_O molecules and OH^-^ ions can accelerated the dissolution of alloying elements, especially Ni, Al, Ti, Cr, W and Ta. The principle reactions included:n/2M + n/2H_2_O → n/2MO + nH^+^ + ne^−^(1)

In which, M stands for metal elements including Ni, Al, Ti, Cr, W, and Ta. The produced H^+^ ions entered the solution and the discharged anodic ions replaced the positions of the O^2−^ ions on sample surface. Or the active dissolution reactions of the alloying elements (Ni, Al, Ti, Cr, and Ta) occurred simultaneously according to:M → M^n+^ + ne^−^(2)

Then the metal ions (Al^3+^, Ti^4+^, Cr^3+^ and Ta^5+^ etc.) reacted with the O^2−^ ions to form metallic oxides. The superalloys which can be resistant to severe corrosion must have a high Cr content, in order to develop a Cr_2_O_3_ protective layer [26,27,28,29]. The effect of Cr on improving the corrosion resistance of nickel-based superalloy is not only the formation of Cr_2_O_3_ is relatively stable, not prone to catastrophic acid fluxing, but also the inhibition and stabilization effect on the formation of NiO [30]. The combination of electrostatic interaction and diffusion motion of molecules and ions caused the redistribution of their positions and the formation of continuous, dense, even, and protective layers. The passive films gave good corrosion resistance for withstanding extreme corrosive environment.

Above 331 mV to 650 mV, the CMSX-4 superalloy went into the transpassivation state, and the current density increased with the increase of the potential again. Some corrosion pits developed in the local areas of the samples at this stage, as shown in Figure 2. The existence of fine particulate matter or the local depletion of alloying elements usually induce the formation of corrosion pits [25,31]. The pits were found to occur randomly at dendrite core and interdendritic region by the aid of SEM, as shown in Figure 3.

The potential went on sweeping above 650 mV at which oxygen anodic evolution occurred. The superalloy suffered general corrosion since then. Numerous pits, voids, and cracks (Figure 4) were generated on the sample surface.

In order to verify the above analysis, three-hour polarization have been tested respectively at five potentials: 50 mV, 200 mV, 250 mV, 600 mV, and 900 mV. Current densities versus time during polarization were recorded in Figure 5, and the initial and the final current densities were listed in Table 1. The 50 mV was the potential at which the current density was very low and the dissolution of metal surface hardly happened. Therefore, no oxides formed on the sample surface. At 200 mV and 250 mV, the final current density pertained to the same order of magnitude as the initial one which evidenced the protective layers on the corroded surface. Of the two, 200 mV was more suitable for the CMSX-4 superalloy to reach the passive state. Their current densities at each potential were very close to the values measured in Figure 1. The current densities at 650 mV and 900 mV were high due to serious corrosion.

### 3.2. Influence of Temperature on the Electrochemical Behaviors of the CMSX-4 Superalloy

Electrochemical parameters that were used to describe the corrosion resistance, such as self-corrosion potential (*E*_corr_), initial passive potential (*E*_pass_), passive current density (*i*_pass_), critical pitting potential (*E*_pit_) at 20 °C, 50 °C, and 80 °C were obtained from Figure 1 and listed in Table 2. The *i*_pass_ was defined as the current density, which fluctuated in a narrow scope when the potential swept the passivation plateau. The *E*_pit_ was defined as the potential at which the current density increased significantly immediately following the stable passivation region. In the polarization test, almost all metal elements were active and initiated pitting when the potential was higher than the pitting potential.

In general, with the temperature increases, the anodic polarization curves moved towards the negative direction, which evidenced the degeneration of the anti-corrosion properties of the CMSX-4 superalloy at higher temperatures. Figure 6 shows the morphologies of corroded samples at 20 °C, 30 °C, 40 °C, 50 °C, and 80 °C, respectively. At 20 °C, no corrosion products were removed from the sample surface. The higher temperature, the more corrosion products, the deeper and larger corrosion pits. Therefore, the corrosion resistance properties of the CMSX-4 superalloy exhibited the sensitivity to the service temperature.

As shown in Figure 4, different regions including the non-peeling tubers A, deciduous position B, and position C were analyzed by using SEM-EDS, as shown in Table 3. The non-peeling tubers which were actually the corrosion products were composed of more O, Ni, Ta, W and less Al, Ti, Cr, Co, Mo. Then to centrifuge and dry the precipitates in the NaCl solution, corrosion products consisted mainly of TaO_2_, WO_3_, and NiO and a small amount of Cr_2_O_5_, CoO, and Al_2_O_3_ by combining the EDS results (Table 4) and the XRD spectra (Figure 7).

### 3.3. Influence of Preoxidation on the Electrochemical Behavior of the CMSX-4 Superalloy

Preoxidation treatment was conducted in air at 900 °C following two procedures and then the oxidized samples were cooled down inside the furnace to room temperature. Heat treatment went through three stages: heating, heat preservation, and cooling. The heating speed, cooling speed, and heat preservation time have been illustrated in Figure 8. The “two procedures” differ from the holding time of heat preservation. The elemental distribution on the surface of the oxidized samples was listed in Table 5. The content of Cr increased obviously with the increase of preoxidation time. SEM results showed that the oxides produced during preoxidation procedure 1 were more uniform, dense, and continuous than that produced during preoxidation procedure 2.

The anodic polarization curves of the CMSX-4 superalloy without and with surface preoxidation (procedure 1 and procedure 2) in 3.5 wt.% NaCl solution are shown in Figure 9. The bare alloy showed a stable passivation stage while the preoxidized alloys showed weak passivity. Moreover, the pitting potential of the bare alloy (331 mV) was more positive than that of the alloy with the preoxidation procedure 2 (317 mV), but more negative than that of the alloy with the preoxidation procedure 1 (442 mV). The alloy with the preoxidation procedure 1 had the best resistance to pitting corrosion.

After peroxidation treatment following procedure 1, a protective oxide layer formed on the sample surface, which improved its corrosion resistance in 3.5 wt.% NaCl solution. However, the oxides formed during preoxidation procedure 2 were porous, leading to degradation of corrosion resistance properties. As shown in Figure 10, the oxide layer spalled significantly after the anodic polarization. Once the oxide layer was damaged, the corrosion rate increased rapidly, accompanied by obvious spallation of corrosion products.

Pitting corrosion is very destructive. When the pitting occurs, the high dissolution rate of the metal is due to the high density of corrosion current flowing over the metal surface. Metal equipment can be perforated in case of serious pitting. Pitting corrosion can also aggravate intergranular corrosion, denudation, stress corrosion cracking, and corrosion fatigue [32,33]. Therefore, for the CMSX-4 superalloy in the paper, once the pitting occurred, the current density rapidly increased and metal surface was destroyed. The failure and spallation of the oxide scales played a more important role for the preoxidized superalloys.

## 4. Conclusions

The electrochemical behaviors of the Ni-base superalloy CMSX-4 have been investigated and the corroded surface and corrosion products have been studied using SEM (equipped with EDS) and XRD. The following conclusions can be yielded:The electrochemical corrosion process contains active dissolution stage, stable passivation stage, transpassivation stage, uniform oxygen evolution stage. The passive film forms within the potential range of 198 mV to 331 mV, causing good corrosion resistance. Pitting corrosion occurs at the transpassivation stage when the potential reaches above 331 mV, becoming the key corrosion mode for the CMSX-4 superalloy.With the increasing temperature, the electrochemical parameters including *E*_corr_, *E*_pass_, *i*_pass_, and *E*_pit_ shift in the negative direction. The higher temperature induces worse corrosion performance.Corrosion products contain NiO, TaO_2_, WO_3_ and other oxides by the EDS and XRD analysis.The failure and spallation of oxide scales becomes the key failure mode for the preoxidized alloys.

## Figures and Tables

**Figure 1 materials-13-05478-f001:**
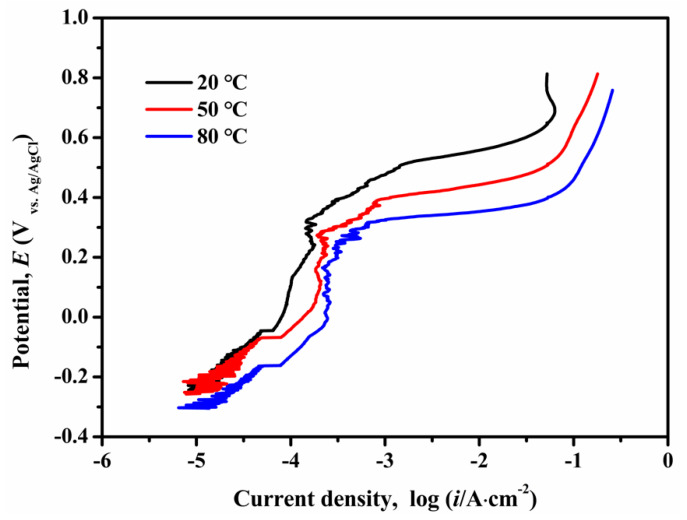
Anodic polarization curves of the CMSX-4 superalloy obtained at 20 °C, 50 °C, and 80 °C in 3.5 wt.% NaCl solution.

**Figure 2 materials-13-05478-f002:**
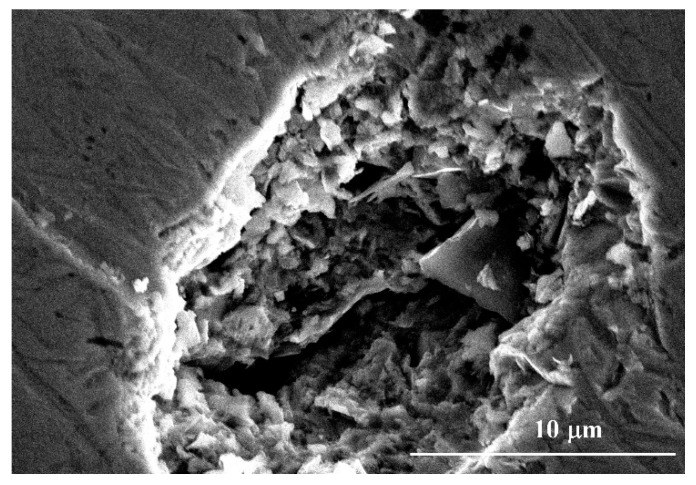
Morphology of the corrosion pit when the potential reaches above 331 mV.

**Figure 3 materials-13-05478-f003:**
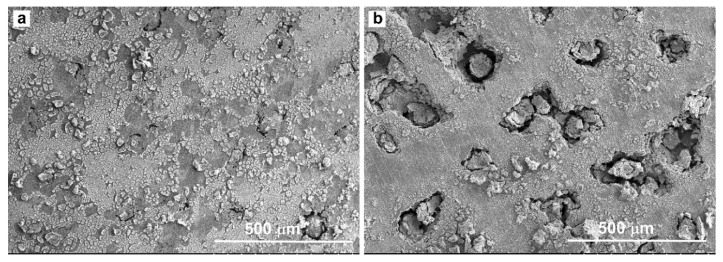
Morphology of pits on the corroded surface. (**a**): Initial stage of pits formation; (**b**): Final stage of pits formation.

**Figure 4 materials-13-05478-f004:**
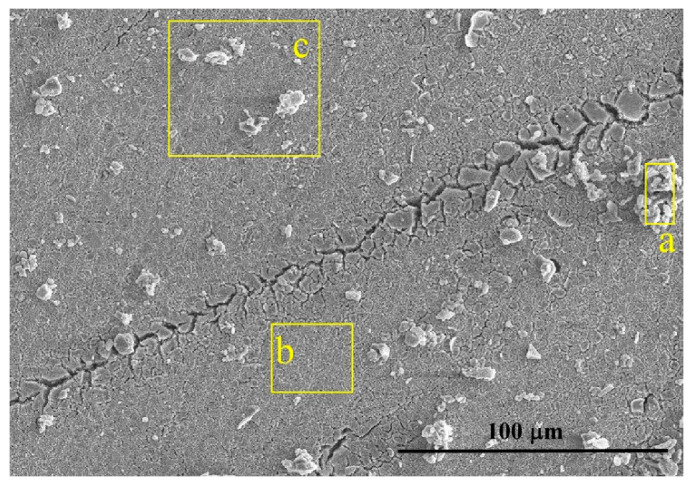
Morphology of the corroded surface when the potential reaches above 650mV. a—non-peeling tubers, b—deciduous position, c—area including non-peeling tubers and deciduous position.

**Figure 5 materials-13-05478-f005:**
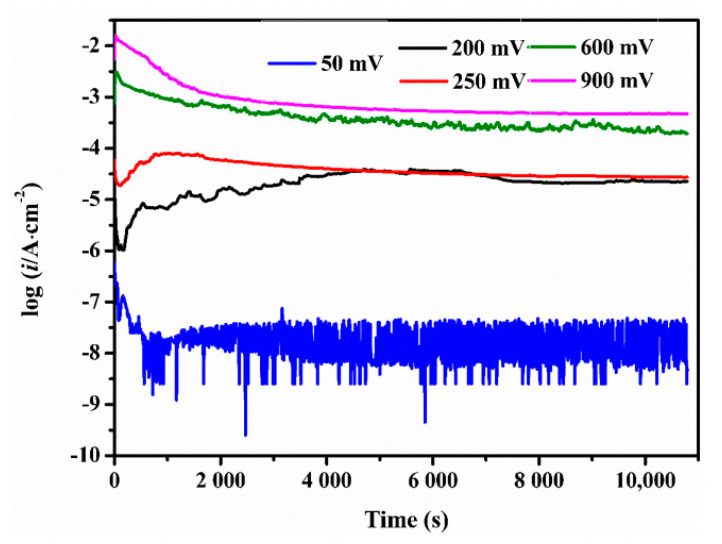
Current densities of the CMSX-4 superalloy during polarization at 50 mV, 200 mV, 250 mV, 600 mV, and 900 mV for 3 h.

**Figure 6 materials-13-05478-f006:**
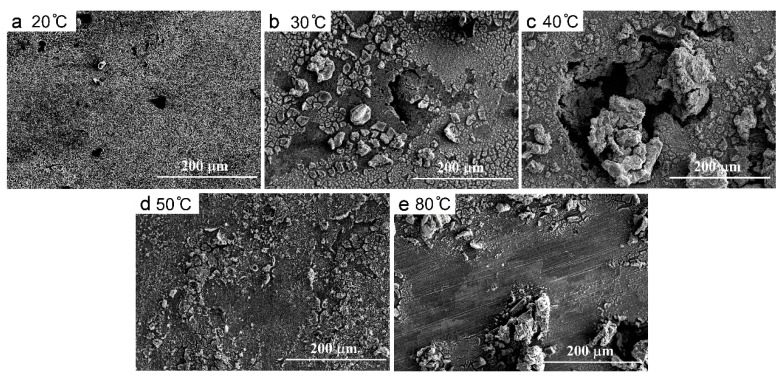
Morphologies of the corroded surfaces at 20 °C (**a**), 30 °C (**b**), 40 °C (**c**), 50 °C (**d**), and 80 °C (**e**).

**Figure 7 materials-13-05478-f007:**
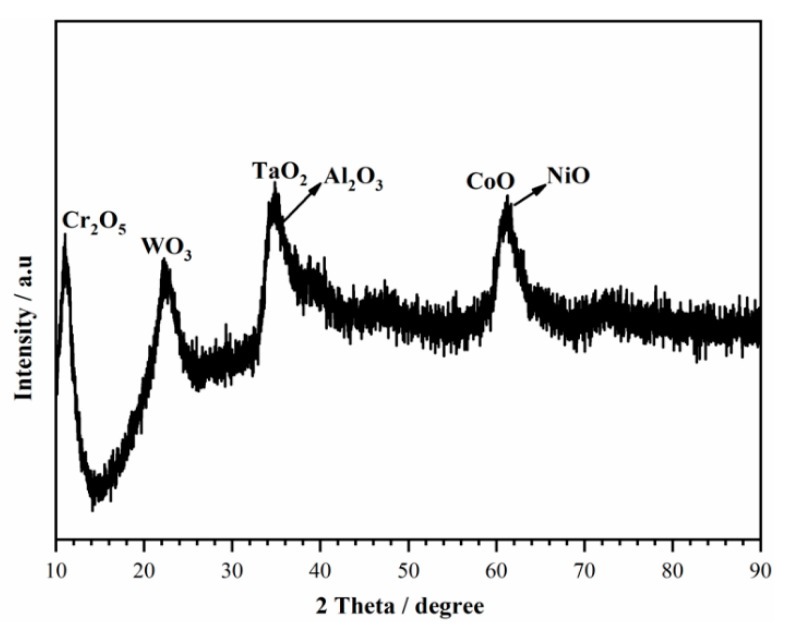
XRD spectra of corrosion products.

**Figure 8 materials-13-05478-f008:**
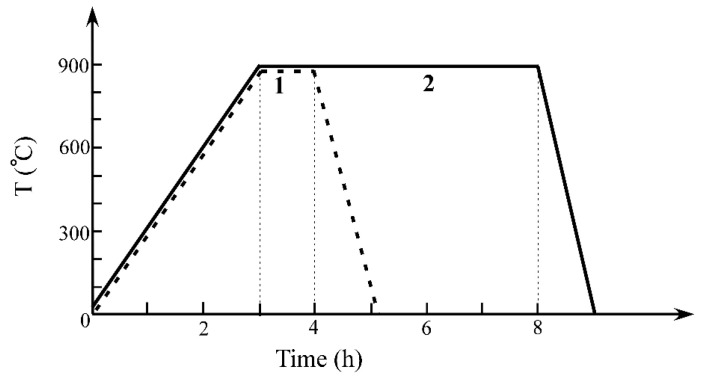
Preoxidation procedures of the CMSX-4 superalloy (temperature vs time), and procedure 1 with 1 h heat preservation, and procedure 2 with 5 h heat preservation.

**Figure 9 materials-13-05478-f009:**
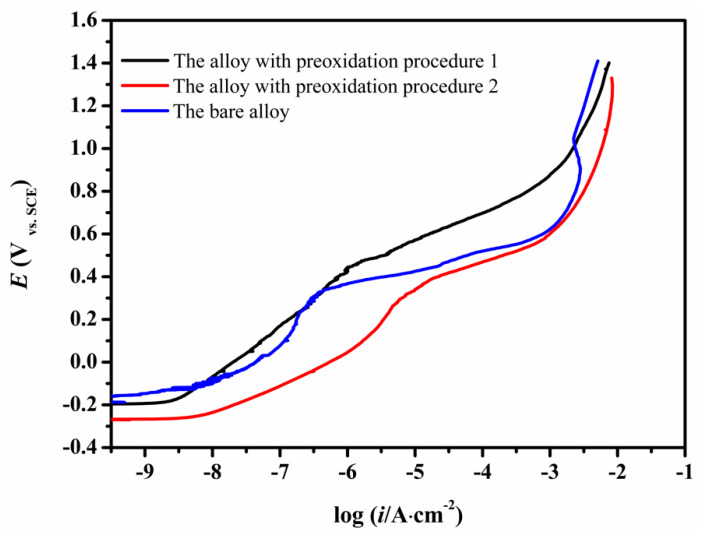
Anodic polarization curves of the bare CMSX-4 sample and the samples obtained following the preoxidation procedures (1 and 2).

**Figure 10 materials-13-05478-f010:**
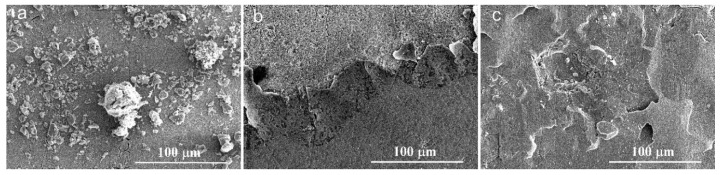
Morphologies of the corroded regions after the anodic polarisation. (**a**) The bare alloy; (**b**) The alloy with preoxidation procedure 1; (**c**) The alloy with preoxidation procedure 2.

**Table 1 materials-13-05478-t001:** The initial and final current densities during 3-h polarizations.

E vs. SCE (mV)	Initial i (μA/cm^2^)	Final i (μA/cm^2^)
50	0.74	0.006
200	20.14	28.97
250	43.35	17.22
600	963.25	248.92
900	7021.66	603.39

**Table 2 materials-13-05478-t002:** Electrochemical parameters obtained from polarization curves at 20 °C, 50 °C, and 80 °C (the E value referred to the Ag/AgCl electrode).

Temperature	*E*_corr_ (mV)	*E*_pass_ (mV)	*i*_pass_ (μA/cm^2^)	*E*_pit_ (mV)
20 °C	−285	198	7.27	331
50 °C	−301	43	7.87	288
80 °C	−325	−10	11.55	169

**Table 3 materials-13-05478-t003:** EDS results of the sample surface after anodic polarization at 20 °C.

	O	Al	Ti	Cr	Co	Ni	Mo	Ta	W	Re
A	32.5	6.3	4.2	4.1	2.6	18.8	2.1	16.8	11.98	0.3
B	8.5	6.7	1.8	5.2	6.6	49.3	1.2	10.8	9.1	0.8
C	10.94	7.21	1.92	4.28	4.333	43.382	1.73	11.34	13.238	0

**Table 4 materials-13-05478-t004:** EDS results of the corrosion products in the NaCl solution.

	Ta	Ni	O	Co	Cr	Cl	Al	W	Mo
Content (wt.%)	80.93	15.04	1.694	1.414	0.307	0.27	0.171	0.08	0.049

**Table 5 materials-13-05478-t005:** EDS results of the sample surface after procedure 1 and procedure 2.

Preoxidation	O	Al	Ti	Cr	Co	Ni	Mo	Ta	W	Re
Procedure 1	19.5	8.0	1.31	9.42	7.28	41.8	0.49	4.04	2.138	0.941
Procedure 2	24.2	7.03	1.9	11.63	7.34	40.83	0.32	4.40	0.646	0.583

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
