# Peer review of "Electrochemical Behaviors of Ni-Base Superalloy CMSX-4 in 3.5 wt.% NaCl: Effect of Temperature and Preoxidation"

_materials, 2020, doi:10.3390/ma13235478_

Round 1

Reviewer 1 Report

The manuscript reports on the effects of temperature and preoxidation treatments on the electrochemical behavior of alloy CMSX-4. Although the scope can be regarded adequate for Materials, the paper must be subjected to a major revision and the authors invited for resubmission when they address the objections and observations listed below.

(1) The main objection to the work as it has been presented regards the discussion on the processes occurring at the surface of the metal as to account for the observed electrochemical behavior. Surface analysis was conducted solely by SEM and EDS, although wrongly stated in Conclusion 3 (line 230) by XRD. In this way, only elemental analysis of the alloy surface can be provided, thus being insufficient to characterize the chemical species present in the surface.

(2) Furthermore, the discussion of the results is mostly related to Cr and Al (see lines 110-115), effectively ignoring all the other components in the alloy, particularly the main nickel component.

(3) How can it be considered WCl6 to be volatile? (line 174)

(4) Results are described in the text without showing any evidence. Namely,

4.1.Lines 126-127 - "The pits was found to occur randomly at dendrite core and interdendritic region by the aid of SEM". NOT SHOWN

4.2. Line 173, "mainly of TaO2 and NiO by combining the EDS results in Table 4 and the XRD spectra". No XRD spectra shown, neither mention to the technique in the experimental section.

4.3.Lines 174-180, How can oxides of these metals react with NaCl? There is no NaCl, but an aqueous phase containing Na+ and Cl- ions!

4.4. Line 188-189, "The oxides on the preoxidized surface included ...". Impossible to ascertain from EDS.

Further comments and recommendations:

5. Include scan rate in the legend of Figure 1. The figure legends should be as much self-explanatory as possible.

6. Potential values must be always referred to a reference. Revise axes in Figures 1 and 7

7. Explain the polarization procedure applied for obtaining the samples imaged by SEM in Figures 2,3,5 and 7. Potential was applied as a stationary potential and for how long? Were they extracted after the application of a scan rate up to the chosen potential?

Experiments must be prompt to replication in another laboratory!

8. Avoid the uses of "too" as it is imprecise. Compared to what? For what? (cf. lines 138, 143, etc.

Author Response

Thank you for the useful comments and suggestions on the images and structure of our manuscript. We have modified the manuscript accordingly, and detailed corrections are listed below point by point:

(1)The main objection to the work as it has been presented regards the discussion on the processes occurring at the surface of the metal as to account for the observed electrochemical behavior. Surface analysis was conducted solely by SEM and EDS, although wrongly stated in Conclusion 3 (line 230) by XRD. In this way, only elemental analysis of the alloy surface can be provided, thus being insufficient to characterize the chemical species present in the surface.

The types of corrosion products can be obtained by XRD analysis. We offered the XRD results in the manuscript (the new Figure 7).

(2) Furthermore, the discussion of the results is mostly related to Cr and Al (see lines 110-115), effectively ignoring all the other components in the alloy, particularly the main nickel component.

For the Eqs. (1)-(4), we chose Cr and Al as the examples in this paper. Actually, all elements including Ni, Al, Cr, Ta, W can take part in the actual reactions. We modified the eqs (1)-(4) using M to stand for mental elements in the new manuscript.

(3) How can it be considered WCl6 to be volatile? (line 174)

Corrosion products in the NaCl solution after centrifuging and drying the precipitates consisted mainly of Ta and Ni in Table 4. In the other word, corrosion products on the sample surface after electrochemical corrosion contain Ni, Ta, W and very small amount of other oxides by the EDS in Table 3. So, we attribute its disappearance to the volatility of WCl6 according to previous report.

But we can not find direct evidence. After email with the editor, we decided to delete this description. This does not affect the integrity of the article and the rationality of the conclusions.

(4) Results are described in the text without showing any evidence. Namely,

For the case of the alloy in this paper electrochemical corroded beneath the 3.5wt.% NaCl solution, the corrosion mechanism could be interpreted as the following model. Cl- could accelerate the migration process of metal oxides through the surface metal oxides and chlorides deposit into the NaCl solution, and thus accelerate corrosion. NaCl would be a very strong corrosive agent. For NaCl, Na+ has and Cl- has only reducibility. Neither the Na+ nor the Cl- could oxidize any components in the superalloy such as Cr, Co, Ni, W et al. So NaCl would probably accelerate the corrosion process, making the corrosion very fast. On the other hand, NaCl in the solution could accelerate the step of metal oxides fluxing into the surface deposit through the Eq. 5 and Eq. 6 (eg, WO3)

One of the products (Na2WO4) could increase alkalinity of the solution and accelerate fluxing of metal oxides into the solution. And another product (chlorides) also migrated to the solution where it re-precipitated into metal oxides following the Eq.7 in this paper:As a result, the Eq. 7 could accelerate the migration of metals from the superalloy surface into the solution. The surface corrosion products (metal oxides including NiO, WO3 et al.) were porous and easily exfoliated.

The reactions occurred under the electrochemical tests would be extremely complicated. Most of all, we can not find direct evidence or calculate the Gibbs free energy to prove the occurrence of Eqs. 5-7.

Therefore, after email with the editor, we decided to delete the Eqs. 5-7, related descriptions, and some controversial assumptions for strictness reasons as the revised version. This does not affect the integrity of the article and the rationality of the conclusions.

4.1.Lines 126-127 - "The pits was found to occur randomly at dendrite core and interdendritic region by the aid of SEM". NOT SHOWN

In this paper, we observed the distribution of pits in different parts. There is no obvious segregation at dendrite core and interdendritic region. We can provide another SEM photo in the manuscript to prove this point (the new Fig 3).

4.2. Line 173, "mainly of TaO2 and NiO by combining the EDS results in Table 4 and the XRD spectra". No XRD spectra shown, neither mention to the technique in the experimental section.

From Table 4, the corrosion products in the NaCl solution mainly included Ta and Ni after centrifuging and drying the precipitates. In fact, parts of the corrosion products were lost during the centrifugation process. The types of the corrosion products were anslyzed in the XRD spectra (Figure 7) in the manuscript.

The description about XRD spectra is shown in “Specimen characterization” section (Double underline highlighted).

4.3.Lines 174-180, How can oxides of these metals react with NaCl? There is no NaCl, but an aqueous phase containing Na+ and Cl- ions!

We made a mistake here. The possible reactions can shown as follows according to several papers:

MO + 2Cl- → MCl2(dissolving) + O2-

MCl2(dissolving) + 1/2 O2 → MO + Cl2

In which, MO stands for metal oxides (M = Co, Cr, and Ni).

[1] Shinata, Y.; Nishi, Y.; NaCl-induced accelerated oxidation of chromium. Oxid. Met. 1986, 26, 201–212.

[2] Y. S. Li. High temperature oxidation and chlorination of metal materials, Doctoral Thesis, Dalian Institute of Science and Technology, Dalian, China, 2001.

[3] Zhang, K.; Liu, M.M.; Liu, S.L.; Sun, C.; Wang, F.H. Hot corrosion behavior of a cobalt-base super-alloy K40S with and without NiCrAlYSi coating. Corros. Sci. 2011, 53, 1990–1998.

4.4. Line 188-189, "The oxides on the preoxidized surface included ...". Impossible to ascertain from EDS.

Due to the long time of sample preparation, part of XRD data was lost. Also, time for modification is limited (10 days) so that we are unable to repeat the experiment. Therefore, we decided to delete this sentence and related description temporarily, which will not affect other descriptions and conclusions in this paper. In the following research, we will improve this experiment if given enough time.

5. Include scan rate in the legend of Figure 1. The figure legends should be as much self-explanatory as possible.

The potential was applied on the working electrode at a scan rate of 0.167 mV/s starting from -0.1 mV to 1.0 mV with respect to self-corrosion potential.

6. Potential values must be always referred to a reference. Revise axes in Figures 1 and 7

For Fig 1, the polarization tests were carried out using the Ag/AgCl electrode reference electrode in saturated KCl solution. For Fig 7, the polarization tests were carried out using saturated calomel electrode (SCE) reference electrode. The reference electrode has been indicated on the potential axis in Fig 1 and Fig 7, respectively.

7. Explain the polarization procedure applied for obtaining the samples imaged by SEM in Figures 2,3,5 and 7. Potential was applied as a stationary potential and for how long? Were they extracted after the application of a scan rate up to the chosen potential?

Fig 2 was obtained when the potential was applied at a scan rate of 0.167 mV/s starting from -0.1 mV to 331mV with respect to self-corrosion potential. Fig 3 was obtained when the potential was applied at a scan rate of 0.167 mV/s starting from -0.1 mV to 650 mV with respect to self-corrosion potential.

If necessary, we can offer the SEM photos when potential was applied as a stationary potential for 3h.

Fig 5 and Fig 7 were obtained after the whole anodic polarization.

8. Avoid the uses of "too" as it is imprecise. Compared to what? For what? (cf. lines 138, 143, etc.

This is a good suggestion.

For the line 138, the 50 mV was the potential at which the current density was very low and the dissolution of metal surface hardly happened. Therefore, no oxides formed on the sample surface.

For the line 143, we delete the “too” to make the statement more accurate.

Reviewer 2 Report

The paper evaluates the effect of high temperature surface pretreatment as well as increased measurement temperature on the corrosion behavior of CMSX-4 superalloy in 3.5% NaCl solution. The topic is interesting and important, the experiments performed are relevant, but a number of technical and formal weaknesses need to be addressed before publication. The conclusions are not well supported (e.g. lack of XRD data) in some cases.

  • The sentence of “Electrochemical dissolution behavior of Nickel-based superalloy revealed that the metallic surface was susceptible to corrosion at low current densities [17]” is not really understandable.
  • Ref 9 and 10 mentions specific alloy high temperature corrosion, while 11-13 takes a general statement on high temperature corrosion of superalloys. Either 11-13 should be specified as well, or 9-10 should be merged in the general statement.
  • “Though the CMSX-4 superalloy has high corrosion resistance, some issues including pitting 57 corrosion, intergranular corrosion, and stress corrosion are studied in a very few papers.” : The authors should mention some of the few references
  • “Though the corrosion of the CMSX-4 superalloy is an electrochemical process, its electrochemical corrosion behaviors are almost researched at low temperature and pressure” : The sentence should be improved grammatically.
  • Materials and Specimen Preparation: Source of the metal sample should be included,size of the electrode and grinding parameters should be included
  • Fig 1.: reference electrode should be indicated on the potential axis
  • The mechanistic explanation (line 109-118) lacks the reference. It is not clear that references 23-26 is about the advantage of chromium content or other mechanistic details.
  • “And the 331 mV was the potential at which pitting corrosion began to happen.” – this sentence is a repetition of the statement in the previous sentence
  • “The 50 mV was too cathodic at which almost no oxide was present on sample surface” – It should b ementioned why it was too cathodic. Compared to what expectations?
  • “The curve fluctuated sharply showing the continual dissolution of metals” – When log I Is around such a a low value then we cannot really call it dissolution.
  • Line 151: „self corrosion” is a strange expression
  • Line 185: „Preoxidation treatment was conducted in air at 900︒C following two procedures and then the oxidized samples were cooled down inside the furnace to room temperature. “ – not well understandable – two types of procedures?
  • Line 209 : “From the Figure 8, during the corrosion process, the oxide layer degraded gradually” – sentence should be improved grammatically
  • Line 212: „Electrochemical corrosion behaviors were controlled by pitting for the bare CMSX-4 superalloy, but  the failure and spallation of the oxide scales played a more important role for the preoxidized superalloys.” – It is not clear how corrosion controls pitting, probably pitting is controlled?

Author Response

Thank you for the useful comments and suggestions on the images and structure of our manuscript. We have modified the manuscript accordingly, and detailed corrections are listed below point by point:

  • The sentence of “Electrochemical dissolution behavior of Nickel-based superalloy revealed that the metallic surface was susceptible to corrosion at low current densities [17]” is not really understandable.

In Paper 17, the electrochemical dissolution behavior of wrought HX (a kind of Nickel-base superalloy) at low current densities was systematically analyzed. The results revealed that M23C6 carbides were irregularly distributed on the grain boundaries, and selective corrosion occurred preferentially on the grain boundary or near the M23C6 precipitations after passivation film polarization. Therefore, this kind of metallic surface was susceptible to corrosion at low current densities. In order to make this concept clear, we will modify this sentence of “Electrochemical dissolution behavior of Nickel-based superalloy revealed that the metallic surface was susceptible to corrosion at low current densities [17]” into “Electrochemical dissolution behavior of Nickel-based superalloy revealed that the presence of M23C6 carbides in the grain boundaries lead to the formation of a number of preferred sites for micro-corrosion cracks at low current densities”.

  • Ref 9 and 10 mentions specific alloy high temperature corrosion, while 11-13 takes a general statement on high temperature corrosion of superalloys. Either 11-13 should be specified as well, or 9-10 should be merged in the general statement.

The multi-layered oxides are very widespread during the high temperature oxidation process though the structure and composition of the layers are significantly different. Therefore, the references 9-10 should be merged in the general statement.

  • “Though the CMSX-4 superalloy has high corrosion resistance, some issues including pitting corrosion, intergranular corrosion, and stress corrosion are studied in a very few papers.” : The authors should mention some of the few references.

We have mentioned and sorted some references about the pitting corrosion, intergranular corrosion and stress corrosion of the CMSX-4 superalloy as following.

[20] Brooking, L.; Gray, S.; Sumner, J.; Nicholls, J.R.; Marchant, G.; Simms, N.J. Effect of stress state and simultaneous hot corrosion on the crack propagation and fatigue life of single crystal superalloy CMSX-4. Int. J. Fatigue 2018, 116, 106–117.

[5] Lortrakul, P.; Trice, R.W.; Trumble, K.P.; Dayananda, M.A. Investigation of the mechanisms of Type-II hot corrosion of superalloy CMSX-4. Corros. Sci. 2014, 80, 408–415.

[21] Zhang, Y.H.; Knowles, D.M.; Withers, P.J. Microstructural development in Pt-aluminide coating on CMSX-4 superalloy during TMF. Surf. Coat. Technol. 1998, 107, 76–83.

[22] Brooking, L.; Sumner, J.; Gray, S.; Simms, N. J. Stress corrosion of Ni-based superalloys. Materials at High Temperatures 2018, 35, 120–129.

[23] Orosz, R.; Krupp, U.; Christ, H.-J.; Monceau, D. The influence of specimen thickness on the high temperature corrosion behavior of CMSX-4 during thermal-cycling exposure. Oxid. Met. 2007, 68, 165–176.

  • “Though the corrosion of the CMSX-4 superalloy is an electrochemical process, its electrochemical corrosion behaviors are almost researched at low temperature and pressure” : The sentence should be improved grammatically.

This sentence have been improved as follows:

Practically, the corrosion process of the CMSX-4 superalloy is an electrochemical process during service. Almost papers about the electrochemical corrosion behaviors of the CMSX-4 superalloy focused on low temperature and pressure.

  • Materials and Specimen Preparation: Source of the metal sample should be included,size of the electrode and grinding parameters should be included

The CMSX-4 superalloy was prepared by State Key Laboratory of Powder Metallurgy (Central South University). The ingots of the alloys were prepared by vacuum induction melting (VIM) and the compositions were measured by inductively-coupled plasma-optical emission spectroscopy (ICP-OES) at NCS Testing Technology Co., Ltd, China. Single crystal rods, with 13mm in diameter and 170mm in length, were directionally solidified using a conventional Bridgman method in a high rate solidification furnace (ALD furnace). The melting temperature was 1500 ℃ and the withdrawal rate of the mold was 3.0 mm/min.

Cylindrical electrodes for electrochemical tests with a diameter of 13 mm and a thickness of 3 mm were prepared by mechanical machining and grinding. Before the tests, the surface of the samples was polished with sandpaper (600#, 1000#, 2000#), then polished to the mirror with Al2O3 polishing paste (1.0μm).

  • Fig 1.: reference electrode should be indicated on the potential axis

The anodic polarization tests were carried out using the Ag/AgCl electrode reference electrode in saturated KCl solution. The reference electrode has been indicated on the potential axis in Fig 1.

  • The mechanistic explanation (line 109-118) lacks the reference. It is not clear that references 23-26 is about the advantage of chromium content or other mechanistic details.

The references 23-26 all indicate that the oxide of Cr has obvious protective effect at the initial stage of corrosion. The effect of Cr content on the corrosion properties of Nickel-base superalloy is another important topic. Some papers represented the advantage of Cr content and its mechanistic details. The effect of Cr on improving the corrosion resistance of nickel-based superalloy is not only the formation of Cr2O3 is relatively stable, not prone to catastrophic acid fluxing, but also the inhibition and stabilization effect on the formation of NiO [31].

[31] Chen, Z.H.; Dong, T.; Qu, W.W.; Ru, Y.; Zhang, H.; Pei, Y.L.; Gong, S.K.; Li, S.S. Influence of Cr content on hot corrosion and a special tube sealing test of single crystal nickel base superalloy. Corros. Sci. 2019, 156, 161–170.

  • “And the 331 mV was the potential at which pitting corrosion began to happen.” – this sentence is a repetition of the statement in the previous sentence

This sentence has been deleted in the manuscript.

  • “The 50 mV was too cathodic at which almost no oxide was present on sample surface” – It should b ementioned why it was too cathodic. Compared to what expectations?

The 50 mV was the potential at which the current density was very low and the dissolution of metal surface hardly happened. Therefore, no oxides formed on the sample surface.

  • “The curve fluctuated sharply showing the continual dissolution of metals” – When log I Is around such a low value then we cannot really call it dissolution.

Sorry, we made this mistake. Practically, the current density fluctuated sharply but remained at a very low value, so the dissolution of metal surface hardly happened.

  • Line 151: „self corrosion” is a strange expression

Some reports used the “self-corrosion potential” to describe the potential measured in the absence of applied current in a particular corrosion system. After the open circuit potential stabilizes, the value of the potential is defined as the “self-corrosion potential”. When we did the polarization test, we applied a scan from negative to positive vs a stable electrode system. However, When the electrode system was disturbed, the “self-corrosion potential” drifted, so the corrosion potential obtained on the polarization diagram was different from the open-circuit potential at the beginning.

  • Line 185: „Preoxidation treatment was conducted in air at 900︒C following two procedures and then the oxidized samples were cooled down inside the furnace to room temperature. “ – not well understandable – two types of procedures?

Heat treatment went through three stages: heating, heat preservation and cooling. The heating speed, the cooling speed and the heat preservation time have been illustrated in Fig 6. The “two procedures” differs from the holding time of heat preservation.

  • Line 209 : “From the Figure 8, during the corrosion process, the oxide layer degraded gradually” – sentence should be improved grammatically

This sentence is modified as follows:

As shown in Figure 8, the oxide layer spalled significantly after the anodic polarization.

  • Line 212: „Electrochemical corrosion behaviors were controlled by pitting for the bare CMSX-4 superalloy, but the failure and spallation of the oxide scales played a more important role for the preoxidized superalloys.” – It is not clear how corrosion controls pitting, probably pitting is controlled?

The original description here is not very clear, now we give an explanation based on several papers.

Pitting corrosion is very destructive. When the pitting occurs, the high dissolution rate of the metal is due to the high density of corrosion current flowing over the metal surface. And then metal equipment can be perforated in case of serious pitting. Pitting corrosion can also aggravate intergranular corrosion, denudation, stress corrosion cracking and corrosion fatigue [32, 33]. Therefore, for the CMSX-4 superalloy in the paper, once the pitting occurred, the current density rapidly increased and metal surface was destroyed.

[32] Brewick, P.T.; Kota, N.; Lewis, A.C.; DeGiorgi, V.G.; Geltmacher, A.B.; Qidwai, S.M. Microstructure-sensitive modeling of pitting corrosion: Effect of the crystallographic orientation. Corros. Sci. 2017, 129, 54-69.

[33] Zhang, Z.; Obasi, G.; Morana, R.; Preuss, M. Hydrogen assisted crack initiation and propagation in a nickel-based superalloy. Acta Mater. 2016, 113, 272-283.

Round 2

Reviewer 1 Report

The authors have made a significant effort to take in account the suggestions and concerns raised by this reviewer, and as result the manuscript is clearer and more precise. Yet, a number of mostly minor errors still require revision, as listed next:

(1) The figures containing potential scales, namely figures 1 and 7, have been edited to contain the reference electrode employed. But the use of two different reference electrodes in each case poses uncertainty regarding all the potential values included in the manuscript and the tables. Therefore, all potential values must be referred to the same reference electrode, and the values measured with respect to the other corrected accordingly to account for the potential difference between the two references.

(2) Following up to item (1), potential values in the Tables must also contain the units referred to the reference electrode as in the Figures.

(3) The anodic polarization curves in Figure 1 have a potential span greatly in excess of 1 mV, in contradiction to what is said in line 110. Probably "V" instead of "mV" were meant to be used.

(4) Line 65 - Replace "almost" with "most"

(5) Line 85 - The surface flattening using sandpaper cannot be named "polishing" but either "abrading" or "grinding"

(6) Line 126 - Ionic charges must be in superscripts

(7) Line 129 - Correct "mental"

(8) Line 196 - There are no various images in Figure 4, but in Figure 6

(9) Line 201 - The plural of "spectrum" is "spectra"

(10) Line 210 - The plural is "differ"

Author Response

Thank you for the useful comments and suggestions on the images and structure of our manuscript. We have modified the manuscript accordingly, and detailed corrections are listed below point by point:

  • The figures containing potential scales, namely figures 1 and 7, have been edited to contain the reference electrode employed. But the use of two different reference electrodes in each case poses uncertainty regarding all the potential values included in the manuscript and the tables. Therefore, all potential values must be referred to the same reference electrode, and the values measured with respect to the other corrected accordingly to account for the potential difference between the two references.

On this issue, we will give an explanation.

First of all, saturated calomel electrode (SCE) will become unstable and seriously damaged at 60 ℃ or higher temperature. Therefore, we use Ag/AgCl electrode to study the effect of temperature on the electrochemical polarization of the CMSX-4 alloy.

Secondly, we referred to the same Ag/AgCl electrode when we obtained all values in Table 2. If these values are corrected with respect to SCE, it will be another complicated problem. Because the standard potential of the two kinds of electrodes is different at different temperatures, even the saturated calomel electrode has been destroyed at higher temperature.

Thirdly, Figures 1 and 7 exist independently. When we discussed Table 2, we also payed attention to the comparison of various indexes at different temperatures. We strictly limited the experimental conditions when we obtained each curve, such as the same instrument, the same reference electrode, the same salt bridge, the same scanning speed and so on. The values obtained are different due to the change of any experimental condition, so there was no emphasis on each value.

  • Following up to item (1), potential values in the Tables must also contain the units referred to the reference electrode as in the Figures.

The reference electrode has been indicated for the potential values in Table 1 and Table 2.

  • The anodic polarization curves in Figure 1 have a potential span greatly in excess of 1 mV, in contradiction to what is said in line 110. Probably "V" instead of "mV" were meant to be used.

This mistake has been corrected in the manuscript.

  • Line 65 - Replace "almost" with "most"

This mistake has been corrected in the manuscript.

  • Line 85 - The surface flattening using sandpaper cannot be named "polishing" but either "abrading" or "grinding"

This mistake has been corrected in the manuscript.

  • Line 126 - Ionic charges must be in superscripts

This mistake has been corrected in the manuscript.

  • Line 129 - Correct "mental"

We have corrected “mental” to “metal” in the manuscript.

  • Line 196 - There are no various images in Figure 4, but in Figure 6

In this sentence, there are various regions (A, B, C) in Figure 4, but various images.

  • Line 201 - The plural of "spectrum" is "spectra"

This mistake has been corrected in the manuscript.

(10) Line 210 - The plural is "differ"

This mistake has been corrected in the manuscript.

Reviewer 2 Report

Two remarks:

  • Fig 5. : "mV" units should be written in the figure as well after the numbers
  • Table 3. : "wt%" should be indicated somehow, at the moment only numbers are given

Author Response

Thank you for the useful comments and suggestions on the images and structure of our manuscript. We have modified the manuscript accordingly, and detailed corrections are listed below point by point:

  • Fig 5. : "mV" units should be written in the figure as well after the numbers

We have added the "mV" units after the numbers in Fig 5.

  • Table 3. : "wt%" should be indicated somehow, at the moment only numbers are given

The table header “content” has been added in Table 3.